# Threshold-Based Overlap of Breast Cancer High-Risk Classification Using Family History, Polygenic Risk Scores, and Traditional Risk Models in 180,398 Women

**DOI:** 10.3390/cancers17213561

**Published:** 2025-11-03

**Authors:** Peh Joo Ho, Christine Kim Yan Loo, Ryan Jak Yang Lim, Meng Huang Goh, Mustapha Abubakar, Thomas U. Ahearn, Irene L. Andrulis, Natalia N. Antonenkova, Kristan J. Aronson, Annelie Augustinsson, Sabine Behrens, Clara Bodelon, Natalia V. Bogdanova, Manjeet K. Bolla, Kristen D. Brantley, Hermann Brenner, Helen Byers, Nicola J. Camp, Jose E. Castelao, Melissa H. Cessna, Jenny Chang-Claude, Stephen J. Chanock, Georgia Chenevix-Trench, Ji-Yeob Choi, Sarah V. Colonna, Kamila Czene, Mary B. Daly, Francoise Derouane, Thilo Dörk, A. Heather Eliassen, Christoph Engel, Mikael Eriksson, D. Gareth Evans, Olivia Fletcher, Lin Fritschi, Manuela Gago-Dominguez, Jeanine M. Genkinger, Willemina R. R. Geurts-Giele, Gord Glendon, Per Hall, Ute Hamann, Cecilia Y. S. Ho, Weang-Kee Ho, Maartje J. Hooning, Reiner Hoppe, Anthony Howell, Keith Humphreys, Hidemi Ito, Motoki Iwasaki, Anna Jakubowska, Helena Jernström, Esther M. John, Nichola Johnson, Daehee Kang, Sung-Won Kim, Cari M. Kitahara, Yon-Dschun Ko, Peter Kraft, Ava Kwong, Diether Lambrechts, Susanna Larsson, Shuai Li, Annika Lindblom, Martha Linet, Jolanta Lissowska, Artitaya Lophatananon, Robert J. MacInnis, Arto Mannermaa, Siranoush Manoukian, Sara Margolin, Keitaro Matsuo, Kyriaki Michailidou, Roger L. Milne, Nur Aishah Mohd Taib, Kenneth R. Muir, Rachel A. Murphy, William G. Newman, Katie M. O’Brien, Nadia Obi, Olufunmilayo I. Olopade, Mihalis I. Panayiotidis, Sue K. Park, Tjoung-Won Park-Simon, Alpa V. Patel, Paolo Peterlongo, Dijana Plaseska-Karanfilska, Katri Pylkäs, Muhammad U. Rashid, Gad Rennert, Juan Rodriguez, Emmanouil Saloustros, Dale P. Sandler, Elinor J. Sawyer, Christopher G. Scott, Shamim Shahi, Xiao-Ou Shu, Katerina Shulman, Jacques Simard, Melissa C. Southey, Jennifer Stone, Jack A. Taylor, Soo-Hwang Teo, Lauren R. Teras, Mary Beth Terry, Diana Torres, Celine M. Vachon, Maxime Van Houdt, Jelle Verhoeven, Clarice R. Weinberg, Alicja Wolk, Taiki Yamaji, Cheng Har Yip, Wei Zheng, Mikael Hartman, Jingmei Li

**Affiliations:** 1Genome Institute of Singapore, Agency for Science, Technology and Research (A*STAR), Singapore 138672, Singapore; 2Saw Swee Hock School of Public Health, National University of Singapore, Singapore 117549, Singapore; 3Division of Cancer Epidemiology and Genetics, National Cancer Institute, National Institutes of Health, Department of Health and Human Services, Bethesda, MD 20850, USA; 4Fred A. Litwin Center for Cancer Genetics, Lunenfeld-Tanenbaum Research Institute of Mount Sinai Hospital, Toronto, ON M5G 1X5, Canada; 5Department of Molecular Genetics, University of Toronto, Toronto, ON M5S 1A8, Canada; 6N.N. Alexandrov Research Institute of Oncology and Medical Radiology, 223040 Minsk, Belarus; 7Department of Public Health Sciences, Sinclair Cancer Research Institute, Queen’s University, Kingston, ON K7L 3N6, Canada; 8Oncology, Department of Clinical Sciences in Lund, Lund University, 221 85 Lund, Sweden; 9Division of Cancer Epidemiology, German Cancer Research Center (DKFZ), 69120 Heidelberg, Germany; 10Department of Population Science, American Cancer Society, Atlanta, GA 30303, USA; 11Department of Radiation Oncology, Hannover Medical School, 30625 Hannover, Germany; 12Gynaecology Research Unit, Hannover Medical School, 30625 Hannover, Germany; 13Centre for Cancer Genetic Epidemiology, Department of Public Health and Primary Care, University of Cambridge, Cambridge CB1 8RN, UK; 14Department of Epidemiology, Harvard T.H. Chan School of Public Health, Boston, MA 02115, USA; 15Dana-Farber Cancer Institute, Department of Medical Oncology, Boston, MA 02215, USA; 16Division of Clinical Epidemiology and Aging Research, German Cancer Research Center (DKFZ), 69120 Heidelberg, Germany; 17German Cancer Consortium (DKTK), German Cancer Research Center (DKFZ), 69120 Heidelberg, Germany; 18Division of Evolution, Infection and Genomics, School of Biological Sciences, University of Manchester, Manchester M13 9PT, UK; 19Department of Internal Medicine and Huntsman Cancer Institute, University of Utah, Salt Lake City, UT 84112, USA; 20Oncology and Genetics Unit, Instituto de Investigación Sanitaria de Santiago de Compostela (IDIS) Foundation, Complejo Hospitalario Universitario de Santiago, SERGAS, 36312 Vigo, Spain; 21Intermountain Healthcare, Salt Lake City, UT 84143, USA; 22Cancer Epidemiology Group, University Cancer Center Hamburg (UCCH), University Medical Center Hamburg-Eppendorf, 20246 Hamburg, Germany; 23Cancer Research Program, QIMR Berghofer Medical Research Institute, Brisbane, QLD 4006, Australia; 24Department of Biomedical Sciences, Seoul National University Graduate School, Seoul 03080, Republic of Korea; 25Cancer Research Institute, Seoul National University, Seoul 03080, Republic of Korea; 26Institute of Health Policy and Management, Seoul National University Medical Research Center, Seoul 03080, Republic of Korea; 27Department of Medical Epidemiology and Biostatistics, Karolinska Institutet, 171 65 Stockholm, Sweden; 28Department of Clinical Genetics, Fox Chase Cancer Center, Philadelphia, PA 19111, USA; 29Leuven Multidisciplinary Breast Center, Department of Oncology, Leuven Cancer Institute, University Hospitals Leuven, 3000 Leuven, Belgium; 30Channing Division of Network Medicine, Department of Medicine, Brigham and Women’s Hospital and Harvard Medical School, Boston, MA 02115, USA; 31Department of Nutrition, Harvard T.H. Chan School of Public Health, Boston, MA 02115, USA; 32Institute for Medical Informatics, Statistics and Epidemiology, University of Leipzig, 04107 Leipzig, Germany; 33LIFE-Leipzig Research Centre for Civilization Diseases, University of Leipzig, 04103 Leipzig, Germany; 34Manchester Centre for Genomic Medicine, St Mary’s Hospital, Manchester University NHS Foundation Trust, Manchester M13 9WL, UK; 35The Breast Cancer Now Toby Robins Research Centre, The Institute of Cancer Research, London SW7 3RP, UK; 36School of Population Health, Curtin University, Perth, WA 6102, Australia; 37Cancer Genetics and Epidemiology Group, Genomic Medicine Group, Fundación Instituto de Investigación Sanitaria de Santiago de Compostela (FIDIS), Complejo Hospitalario Universitario de Santiago, Servicio Gallego de Salud (SERGAS), 15706 Santiago de Compostela, Spain; 38Department of Epidemiology, Mailman School of Public Health, Columbia University, New York, NY 10032, USA; 39Herbert Irving Comprehensive Cancer Center, New York, NY 10032, USA; 40Department of Clinical Genetics, Erasmus University Medical Center, 3015 CN Rotterdam, The Netherlands; 41Department of Oncology, Södersjukhuset, 118 83 Stockholm, Sweden; 42Molecular Genetics of Breast Cancer, German Cancer Research Center (DKFZ), 69120 Heidelberg, Germany; 43Department of Molecular Pathology, Hong Kong Sanatorium and Hospital, Hong Kong, China; 44School of Mathematical Sciences, Faculty of Science and Engineering, University of Nottingham Malaysia, Semenyih, Selangor 43500, Malaysia; 45Cancer Research Malaysia, Subang Jaya, Selangor 47500, Malaysia; 46Department of Medical Oncology, Erasmus MC Cancer Institute, 3015 GD Rotterdam, The Netherlands; 47Dr. Margarete Fischer-Bosch-Institute of Clinical Pharmacology, 70376 Stuttgart, Germany; 48University of Tübingen, 72074 Tübingen, Germany; 49Division of Cancer Sciences, University of Manchester, Manchester M13 9PL, UK; 50Division of Cancer Information and Control, Aichi Cancer Center Research Institute, Nagoya 464-8681, Japan; 51Division of Cancer Epidemiology, Nagoya University Graduate School of Medicine, Nagoya 466-8550, Japan; 52Division of Epidemiology, National Cancer Center Institute for Cancer Control, Tokyo 104-0045, Japan; 53Independent Laboratory of Molecular Biology and Genetic Diagnostics, Pomeranian Medical University, 171-252 Szczecin, Poland; 54International Hereditary Cancer Center, Department of Genetics and Pathology, Pomeranian Medical University, 171-252 Szczecin, Poland; 55Department of Epidemiology and Population Health, Stanford University School of Medicine, Stanford, CA 94305, USA; 56Department of Medicine, Division of Oncology, Stanford Cancer Institute, Stanford University School of Medicine, Stanford, CA 94304, USA; 57Department of Preventive Medicine, Seoul National University College of Medicine, Seoul 03080, Republic of Korea; 58Department of Surgery, Daerim Saint Mary’s Hospital, Seoul 07442, Republic of Korea; 59Radiation Epidemiology Branch, Division of Cancer Epidemiology and Genetics, National Cancer Institute, Bethesda, MD 20892, USA; 60Department of Internal Medicine, Johanniter GmbH Bonn, Johanniter Krankenhaus, 53177 Bonn, Germany; 61Hong Kong Hereditary Breast Cancer Family Registry, Hong Kong, China; 62Department of Surgery, The University of Hong Kong, Hong Kong, China; 63Department of Surgery and Cancer Genetics Center, Hong Kong Sanatorium and Hospital, Hong Kong, China; 64Laboratory for Translational Genetics, Department of Human Genetics, KU Leuven, 3000 Leuven, Belgium; 65VIB Center for Cancer Biology, Vlaams Instituut voor Biotechnologie (VIB), 3001 Leuven, Belgium; 66Institute of Environmental Medicine, Karolinska Institutet, 171 77 Stockholm, Sweden; 67Department of Surgical Sciences, Uppsala University, 751 05 Uppsala, Sweden; 68Centre for Epidemiology and Biostatistics, Melbourne School of Population and Global Health, The University of Melbourne, Melbourne, VIC 3010, Australia; 69Precision Medicine, School of Clinical Sciences at Monash Health, Monash University, Clayton, VIC 3168, Australia; 70Department of Molecular Medicine and Surgery, Karolinska Institutet, 171 76 Stockholm, Sweden; 71Department of Clinical Genetics and Genomics, Karolinska University Hospital, 171 76 Stockholm, Sweden; 72Department of Cancer Epidemiology and Prevention, M. Sklodowska-Curie National Research Oncology Institute, 02-034 Warsaw, Poland; 73Division of Population Health, Health Services Research and Primary Care, School of Health Sciences, Faculty of Biology, Medicine and Health, The University of Manchester, Manchester M13 9PL, UK; 74Cancer Epidemiology Division, Cancer Council Victoria, Melbourne, VIC 3004, Australia; 75Translational Cancer Research Area, University of Eastern Finland, 70210 Kuopio, Finland; 76Institute of Clinical Medicine, Pathology and Forensic Medicine, University of Eastern Finland, 70210 Kuopio, Finland; 77Biobank of Eastern Finland, Kuopio University Hospital, 70210 Kuopio, Finland; 78Unit of Medical Genetics, Department of Medical Oncology and Hematology, Fondazione IRCCS Istituto Nazionale dei Tumori di Milano, 20133 Milan, Italy; 79Department of Clinical Science and Education, Södersjukhuset, Karolinska Institutet, 118 83 Stockholm, Sweden; 80Division of Cancer Epidemiology and Prevention, Aichi Cancer Center Research Institute, Nagoya 464-8681, Japan; 81Biostatistics Unit, The Cyprus Institute of Neurology and Genetics, Nicosia 2371, Cyprus; 82Department of Surgery, Faculty of Medicine, UM Cancer Research Institute, University of Malaya, Kuala Lumpur 50603, Malaysia; 83School of Population and Public Health, University of British Columbia, Vancouver, BC V6T 1Z4, Canada; 84Cancer Control Research, BC Cancer Agency, Vancouver, BC V5Z 1L3, Canada; 85Epidemiology Branch, National Institute of Environmental Health Sciences, National Institutes of Health (NIH), Research Triangle Park, Durham, NC 27709, USA; 86Institute for Occupational and Maritime Medicine, University Medical Center Hamburg-Eppendorf, 20246 Hamburg, Germany; 87Institute for Medical Biometry and Epidemiology, University Medical Center Hamburg-Eppendorf, 20246 Hamburg, Germany; 88Center for Clinical Cancer Genetics, The University of Chicago, Chicago, IL 60637, USA; 89Department of Cancer Genetics, Therapeutics and Ultrastructural Pathology, The Cyprus Institute of Neurology & Genetics, Nicosia 2371, Cyprus; 90Department of Comparative Biomedical Sciences, College of Veterinary Medicine, Mississippi State University, Starkville, MS 39762, USA; 91Integrated Major in Innovative Medical Science, Seoul National University College of Medicine, Seoul 03080, Republic of Korea; 92Laboratory of Hematology-Oncology, European Institute of Oncology IRCCS, 20133 Milan, Italy; 93Research Centre for Genetic Engineering and Biotechnology ‘Georgi D. Efremov’, Macedonian Academy of Sciences and Arts (MASA), 1000 Skopje, North Macedonia; 94Laboratory of Cancer Genetics and Tumor Biology, Translational Medicine Research Unit, Biocenter Oulu, University of Oulu, 90220 Oulu, Finland; 95Laboratory of Cancer Genetics and Tumor Biology, Northern Finland Laboratory Centre, 90220 Oulu, Finland; 96Department of Basic Sciences, Shaukat Khanum Memorial Cancer Hospital and Research Centre (SKMCH & RC), Lahore 54000, Pakistan; 97Faculty of Medicine, Technion-Israel Institute of Technology and Association for Promotion of Research in Precision Medicine, Haifa 35254, Israel; 98Division of Oncology, Faculty of Medicine, School of Health Sciences, University of Thessaly, 411 10 Larissa, Greece; 99School of Cancer & Pharmaceutical Sciences, Comprehensive Cancer Centre, Guy’s Campus, King’s College London, London SE1 1UL, UK; 100Department of Quantitative Health Sciences, Division of Clinical Trials and Biostatistics, Mayo Clinic, Rochester, MN 55905, USA; 101Division of Epidemiology, Department of Medicine, Vanderbilt Epidemiology Center, Vanderbilt-Ingram Cancer Center, Vanderbilt University School of Medicine, Nashville, TN 37203, USA; 102Clalit Regional Oncology Unit, Haifa and Western Galilee District, Haifa 3436212, Israel; 103Genomics Center, Centre Hospitalier Universitaire de Qubec-Universit Laval Research Center, Quebec City, QC G1V 4G2, Canada; 104Department of Clinical Pathology, The University of Melbourne, Melbourne, VIC 3010, Australia; 105Genetic Epidemiology Group, School of Population and Global Health, University of Western Australia, Perth, WA 6000, Australia; 106Epigenetic and Stem Cell Biology Laboratory, National Institute of Environmental Health Sciences, National Institutes of Health (NIH), Research Triangle Park, Durham, NC 27709, USA; 107Institute of Human Genetics, Pontificia Universidad Javeriana, Bogota 110231, Colombia; 108Department of Quantitative Health Sciences, Division of Epidemiology, Mayo Clinic, Rochester, MN 55905, USA; 109Department of Gynecological Oncology, University Hospitals of Leuven, 3000 Leuven, Belgium; 110Biostatistics and Computational Biology Branch, National Institute of Environmental Health Sciences, National Institutes of Health (NIH), Research Triangle Park, Durham, NC 27709, USA; 111Subang Jaya Medical Centre, Subang Jaya, Selangor 47500, Malaysia; 112Department of Surgery, National University Health System, Singapore 119228, Singapore; 113Department of Surgery, Yong Loo Lin School of Medicine, National University of Singapore, Singapore 117600, Singapore; 114National Cancer Centre, Singapore Health Services (SingHealth), Singapore 168583, Singapore; 115Australian Breast Cancer Tissue Bank, Westmead Institute for Medical Research, University of Sydney, Sydney, NSW 2145, Australia; 116Research Department, Peter MacCallum Cancer Center, Melbourne, VIC 3000, Australia; 117Sir Peter MacCallum Department of Oncology, The University of Melbourne, Parkville, VIC 3010, Australia; 118Breast Cancer Research Unit, University Malaya Cancer Research Institute, Faculty of Medicine, University of Malaya, Kuala Lumpur 50603, Malaysia; 119Cancer Genetics Service, National Cancer Centre, Singapore 169610, Singapore; 120Breast Department, KK Women’s and Children’s Hospital, Singapore 229899, Singapore; 121SingHealth Duke-NUS Breast Centre, Singapore 168753, Singapore; 122Department of General Surgery, Tan Tock Seng Hospital, Singapore 308433, Singapore; 123Division of Surgery and Surgical Oncology, National Cancer Centre, Singapore 169610, Singapore; 124Department of General Surgery, Singapore General Hospital, Singapore 169608, Singapore; 125Division of Breast Surgery, Department of General Surgery, Changi General Hospital, Singapore 529889, Singapore; 126Division of Radiation Oncology, National Cancer Centre, Singapore 169610, Singapore; 127Division of Medical Oncology, National Cancer Centre, Singapore 169610, Singapore

**Keywords:** breast cancer, ductal carcinoma in situ (DCIS), polygenic risk score (PRS), Gail model, risk stratification, *BRCA1*, *BRCA2*, risk-based screening

## Abstract

**Simple Summary:**

Breast cancer is influenced by both inherited genetic factors and lifestyle or personal factors such as age, family history, and reproductive history. Scientists have developed tools to estimate a woman’s risk of developing breast cancer. One type of tool, called a polygenic risk score, uses many small genetic variations to estimate risk, while another, the Gail model, uses personal and family medical information. We studied how well these tools predict breast cancer risk in women of European and Asian backgrounds. Our research included more than 180,000 women and compared performance across age groups and cancer types. We found that genetic scores were especially useful in younger women and in women of Asian background, while the Gail model worked better in older women of European background. However, both tools showed some inaccuracy when comparing predicted and observed risks. Overall, combining genetic information with traditional risk factors could improve how doctors identify women at higher risk for breast cancer, leading to more personalized screening and prevention strategies across different populations.

**Abstract:**

**Background**: Breast cancer polygenic risk scores (PRS) and traditional risk models (e.g., the Gail model [Gail]) are known to contribute largely independent information, but it is unclear how the overlap varies by ancestry, age, disease type (invasive breast cancer, DCIS), and risk threshold. **Methods**: In a retrospective case–control study, we evaluated risk prediction performance in 180,398 women (161,849 of European ancestry; 18,549 of Asian ancestry). Odds ratios (ORs) from logistic regression models and the area under the receiver operating characteristic curve (AUC) were estimated. **Results**: PRS for invasive disease showed a stronger association in younger (<50 years) women (OR = 2.51, AUC = 0.622) than in women ≥ 50 years (OR = 2.06, AUC = 0.653) of European ancestry. PRS performance in Asians was lower (OR range = 1.62–1.64, AUC = 0.551–0.600). Gail performance was modest across groups and poor in younger Asian women (OR = 0.94–0.99, AUC = 0.523–0.533). Age interactions were observed for both PRS (*p* < 0.001) and Gail (*p* < 0.001) in Europeans, whereas in Asians, age interaction was observed only for Gail (invasive: *p* < 0.001; DCIS: *p* = 0.002). PRS identified more high-risk individuals than Gail in Asian populations, especially ≥50 years, while Gail identified more in Europeans. Overlap between PRS, Gail, and family history was limited at higher thresholds. Calibration analysis, comparing empirical and model-based ROC curves, showed divergence for both PRS and Gail (*p* < 0.001), which indicates miscalibration. In Europeans, family history and prior biopsies drove Gail discrimination. In younger Asians, age at first live birth was influential. **Conclusions**: PRS adds value to risk stratification beyond traditional tools, especially in younger women and Asian ancestry populations.

## 1. Introduction

Breast cancer risk arises from many factors, including inherited genetic mutations (e.g., *BRCA1/2*), reproductive and hormonal history, and lifestyle exposures [1,2]. However, in everyday clinical practice, age and family history (FH) remain the commonly used predictors for preliminary risk stratification. Polygenic risk scores (PRS) and traditional breast cancer risk models, such as the Gail model (Gail), the Tyrer–Cuzick model, or the Breast Cancer Surveillance Consortium (BCSC) model, have each demonstrated predictive power in prior studies [3]. They have been shown to contribute largely independent information and can identify non-overlapping high-risk individuals when combined with FH and pathogenic variants of breast cancer predisposition genes [3,4,5]. Combining PRS with non-genetic risk factors, including mammographic density, has consistently improved discriminatory performance compared to using a single predictor [3]. Many ongoing trials of risk-based breast cancer screening (e.g., MyPEBS and WISDOM) are already implementing integrated risk models [6,7].

Although integrated models generally improve discrimination, their applicability may be limited in populations where there is a lack of validation data. In this context, evaluating PRS and traditional breast cancer risk models comprising non-genetic risk factors separately enables the quantification of (a) unique genetic risk attributed to PRS, (b) distinctive predictive value from clinical and epidemiologic factors, and (c) how each independently identifies high-risk individuals with insights on potential miscalibration. For example, the BREATHE study in Singapore stratifies women using separate risk domains (i.e., PRS, non-genetic risk factors (Gail), mammographic density, and recall history) and considers women as high risk if they exceed thresholds in any one category [8,9]. This approach reveals the proportion of women uniquely identified by PRS, traditional risk factors, and imaging as high-risk. The level of incomplete risk capture may inform policy decisions in settings where PRS assessment is not yet routine.

Typically, women are stratified into two categories of breast cancer risk (high/not high). A wide range of five-year absolute-risk thresholds (1.3%, corresponding to the risk of an average 50-year-old Caucasian woman, to 3% for intervention eligibility) has been used to define elevated breast cancer risk in academic literature and guidelines [4,5]. The National Comprehensive Cancer Network (NCCN) guidelines specify that women aged ≥35 years, with a life expectancy of at least ten years and a five-year Gail risk ≥ 1.67%, may be considered for risk-reducing therapies [10]. The US Preventive Services Task Force (USPSTF) recommends a higher 5-year risk threshold of ≥3.0% to define elevated risk where the benefits of chemoprevention (tamoxifen or raloxifene) are likely to outweigh harms in most women [11]. As risk thresholds for initiating or escalating surveillance (e.g., imaging by mammography) often differ from those guiding preventive interventions, the same risk cut-off may not apply. To assess how different predictors capture unique high-risk women, multiple threshold values should be evaluated.

The incidence of ductal carcinoma in situ (DCIS), a non-invasive form of breast cancer, has increased markedly in recent decades, and DCIS now accounts for approximately 20–25% of newly diagnosed breast cancers [12]. DCIS is widely regarded as a precursor to invasive breast cancer [13,14]. Not unexpectedly, DCIS and invasive disease share many risk factors [15]. However, both Gail and PRS were developed and validated specifically for predicting invasive breast cancer, not DCIS (Gail excludes women with prior DCIS or LCIS). The transferability of these models to DCIS remains uncertain. As DCIS constitutes a non-negligible healthcare burden in terms of overdiagnosis, adverse treatment-related effects, and costs, and given the shared risk factors between DCIS and invasive breast cancer, it is worthwhile to explore the application of invasive cancer prediction tools for DCIS [16,17].

This study aims to evaluate the performance of genetic (PRS) and non-genetic (Gail) breast cancer risk predictors in identifying high-risk individuals across different age groups (<50 years and ≥50 years) and ancestries (Asian and European). By examining the proportion of individuals identified as high risk by these factors across various risk thresholds (five-year absolute risk 0.5–2.5%), we seek to determine how well each predictor performs in diverse populations and age groups.

## 2. Methods

### 2.1. Study Population

The Breast Cancer Association Consortium (BCAC) is an international collaboration that was formed to provide large sample sizes for investigating genetic associations [18]. Women diagnosed with invasive breast cancer DCIS and cancer-free controls were recruited by study groups globally and collectively studied under BCAC [17]. Our retrospective case–control study focuses on individuals who are genetically Asian or European White (from here on referred to as “European”). Details of the ancestry analysis (the “EthnicityGeno” variable used in BCAC analyses) have been previously described [19].

To reduce the influence of missing values on the performance of Gail, studies were excluded if they had missing values for at least two of the three risk factors in the model (age at menarche, age at first live birth, and first-degree breast cancer FH), for 50% or more of participants [20]. The studies included are listed in Appendix A. Exclusion was determined separately for individual studies and each disease status (invasive, DCIS, and controls).

Further exclusions were made on an individual level (Appendix A). Women with unknown age at enrolment for controls (n = 5566) and unknown age at diagnosis for invasive breast cancer or DCIS (n = 2103) were excluded. Women below the age of 30 years (n = 2360) and above 80 years (n = 1897) of age for whom Gail prediction is not valid were excluded. A total of 180,398 individuals were included in our study. We compared demographic differences between the included and excluded individuals to assess potential selection bias.

### 2.2. Prediction Models

#### 2.2.1. Gail Model

Due to the large number of studies with varying degrees of missing data for different risk factors, the parsimonious Gail model, which most studies would have information on, was selected [20]. The model uses information on reproductive risk factors (age of menarche, age at first live birth), personal history (number of breast biopsies, and history of atypical hyperplasia), and family (first-degree) history of breast cancer. The R package “BCRA” (version 2.1.2) was used to calculate five-year absolute risk. Missing values were recoded to the baseline category by the package (i.e., relative risk = 1). In addition, FH (yes/no) was separately studied. Those with unknown FH were considered to have no FH. Five-year absolute risks were estimated by applying the breast cancer incidence rates and mortality rates of “Whites” and “Chinese” (“BCRA” package) to the European and Asian genetic subgroups, respectively.

#### 2.2.2. Handling of Missing Data and Sensitivity Analyses

Information on the number of prior breast biopsies was unavailable for the majority of participants (94% of included European-ancestry and 100% of included Asian-ancestry participants). Because this variable was almost entirely missing, multiple imputation was not performed, as imputed values would have been determined primarily by model assumptions rather than observed data. For the primary analysis, missing values for all Gail model variables were assigned to the reference (lowest-risk) category, consistent with prior validation studies of the model.

To assess the potential influence of missing data on model performance, sensitivity analysis was conducted in which missing values were instead assigned to the highest-risk category for each variable (age at menarche, age at first live birth, number of first-degree relatives with breast cancer, and number of prior breast biopsies). Five-year absolute risk was calculated using the R package BCRA, and the model’s discriminatory ability for invasive breast cancer was evaluated using the area under the receiver operating characteristic curve (AUC) with 95% confidence intervals.

#### 2.2.3. PRS

Among the multiple PRSs available for breast cancer, we used the 313-variant breast cancer PRS developed by the Breast Cancer Association Consortium (BCAC). This PRS was selected because it has been extensively validated in large-scale studies across diverse populations and has demonstrated strong and consistent associations with breast cancer risk and reproducible discriminatory performance [15,21]. Details of the variants, including allele frequencies stratified by ancestry, age, and disease status in our analytical dataset, are presented in Appendix A.

PRS was calculated with the --score function (scoresum option) in PLINK2 [22]. When PLINK2 encounters missing dosage entries (e.g., NA, -9, or blank), it applies mean imputation to replace the missing value with the allele frequency calculated across the dataset (population average as opposed to zero). The rates from the “BCRA” package were used to maintain comparability between the absolute risk estimated for the PRS and Gail. Details for the calculation of absolute risk were published by Mavaddat et al. [21]. In brief, an individual’s PRS percentile was obtained from the standardized PRS using the “pnorm” function in R. Standardization was performed using the ancestry-specific means and standard deviations of the controls (Appendix A). The five-year absolute risk was calculated by estimating the theoretical odds ratio of this percentile in relation to the 40–60th percentile, which was taken to represent the general population [23].

### 2.3. Statistical Analysis

Differences in characteristics for invasive breast cancer cases, DCIS cases, and controls were assessed using the Chi-squared test (categorical variables) and the Kruskal–Wallis test (continuous variables). We assessed the relationship between estimated five-year absolute breast cancer risk (modeled as a continuous variable using PRS and the Gail model) and invasive breast cancer or DCIS. Logistic regression models were fitted to estimate odds ratios (ORs) and corresponding 95% confidence intervals (CIs). Analyses were stratified by disease (invasive, DCIS), genetic ancestry (Asian, European), and age group (<50 years, ≥50 years). This approach provides directly interpretable, age-specific effect estimates without relying on interaction terms. Formal interaction tests between age (modeled as a continuous variable) and each risk score were conducted separately and reported as P interaction values to assess potential heterogeneity of effects by age. Venn diagrams (R package “VennDiagram”, v1.7.3) were used to ascertain the overlaps in high-risk individuals identified by PRS and Gail for five-year absolute-risk thresholds from 0.5 to 2.5%, and FH (binary) for the different groups. Traditional evaluation of calibration often relies on the Hosmer–Lemeshow goodness-of-fit test, which partitions subjects into risk deciles and compares observed versus predicted events. However, in our case–control study, the observed event rate in the sample does not represent the true population prevalence, making the standard Hosmer–Lemeshow test unreliable without design-specific adjustment. Hence, we used the “model-based” ROC curve approach, which focuses on the relative accuracy of predicted risks [24]. The slope is valid without adjustment, reflecting over- or under-fitting in terms of risk spread. The European-ancestry population aged ≥50 years was set as the reference for comparisons.

Although studies with high missingness rates for variables required to compute the Gail risk score were excluded, there were still individuals with missing values. Hence, we studied the potential drivers of Gail in discriminating invasive breast cancer cases from controls using logistic regression models. All combinations of risk factors, where available, were assessed. Discriminatory ability was assessed by the area under the receiver operator curve (AUC).

All analyses were performed in R (version 4.5.0), unless otherwise stated. All analytical code used in this study is publicly available at the GitHub repository: https://github.com/ryan-limjy/Gene.and.Tonic.

## 3. Results

### 3.1. Excluded Participants

Appendix A compares included and excluded participants (all from studies missing ≥50% of data for at least two of the three Gail variables) by ancestry. Excluded women of European ancestry were younger at interview or diagnosis (mean age of 54 vs. 57; *p* < 0.001), and none had information on prior biopsies. Excluded Asians were particularly likely to lack data on age at menarche, age at first full-term pregnancy, and FH. Excluded European-ancestry women differed significantly from those included regarding age at first full-term pregnancy, and missing data on age at menarche was more common among excluded European-ancestry women. Across all subgroups, excluded individuals had significantly lower five-year absolute risk according to Gail (*p* < 0.001). The PRS sum score was significantly higher in excluded European-ancestry women (−0.237 vs. −0.247; *p* = 0.002) and in excluded Asians (0.308 vs. 0.279; *p* = 0.002). For Asian-ancestry women, excluded individuals also had significantly higher PRS-based five-year absolute risk (0.716 vs. 0.681; *p* < 0.001).

Appendix A further stratify participants by age group (<50 years vs. ≥50 years) and disease status (controls, invasive breast cancer, DCIS). In older European invasive breast cancer cases, excluded participants had higher PRS (−0.083 vs. −0.107; *p* < 0.001). Younger European controls and invasive cases had lower PRS-based five-year risk when excluded. Among Asians, except DCIS, excluded individuals consistently showed higher PRS scores or risk estimates than those included (*p* < 0.05).

### 3.2. Analytical Cohort

A total of 180,398 women were included, where 161,849 (90%) women were of European ancestry (52% invasive and 6% DCIS) and 18,549 (10%) were of Asian ancestry (50% invasive and 5% DCIS) (Table 1).

#### 3.2.1. European Ancestry

The median age at diagnosis for invasive cases of European ancestry was 57 years [interquartile range [IQR]: 49–65]. The corresponding age at interview for European-ancestry controls was 57 years [IQR: 50–64] (Table 1). Invasive cases were more likely to have a FH than controls (14% vs. 9%, respectively). The PRS distributions and corresponding five-year absolute risks were similar across countries (Appendix A). The observations were largely similar for DCIS (Table 1).

#### 3.2.2. Asian Ancestry

The median age at diagnosis for Asian-ancestry invasive cases was younger at 49 years [IQR: 43–57], and the age at enrolment was 50 years [IQR: 44–58] for controls (Table 1). Of the invasive cases, 10% reported positive FH, compared to a smaller proportion of controls (6%). In contrast to the European ancestry group, the distribution of PRS and five-year absolute risks varied by country (Appendix A). As with the Europeans, the observations were mostly consistent for DCIS in those of Asian ancestry (Table 1).

#### 3.2.3. Performance of Risk Models

In invasive breast cancer among individuals of European ancestry, younger women (<50 years) exhibited a stronger PRS association (OR:2.51 [2.39–2.62]) but lower discrimination (AUC: 0.622 [0.617–0.628]), whereas in older women (≥50 years), the PRS effect was weaker (OR: 2.06 [2.02–2.11]) with higher discrimination (AUC: 0.653 [0.650–0.656]) (Table 2). In contrast, for DCIS, younger women showed both stronger association and better discrimination (OR: 2.56 [2.37–2.78]; AUC = 0.657 [0.645–0.669]), while older women had lower OR and AUC (OR: 1.56 [1.51–1.61]; AUC = 0.620 [0.613–0.626]).

In a sensitivity analysis assuming missing values corresponded to the highest risk category, the Gail model’s discriminatory ability changed notably among European-ancestry women. (Appendix A) For invasive disease, the AUC for the full model increased from 0.493 (0.487–0.499) to 0.627 (0.621–0.632) and for women < 50 years and from 0.517 (0.514–0.520) to 0.561 (0.557–0.564) for those ≥50 years, suggesting that the large proportion of missing data, particularly for family history and biopsy variables, may have led to underestimation of model discrimination in the original analysis. Among Asian women, AUCs were similar between the two approaches (<50 years: 0.523 [0.511–0.535] vs. 0.507 [0.494–0.519]; ≥50 years: 0.554 [0.543–0.566] vs. 0.531 [0.520–0.543]).

In contrast, for DCIS, the Gail model showed minimal change or a reduction in AUCs when missing data were assigned to the high-risk category (Appendix A). Among European women, the AUC for the full model was 0.521 (0.509–0.533) compared with 0.610 (0.597–0.622) in the original model for those <50 years, and 0.525 (0.518–0.531) vs. 0.519 (0.512–0.526) for those ≥50 years. Among Asian women, AUCs were similar or modestly higher under the high-risk assumption (0.589 (0.560–0.619) vs. 0.533 (0.505–0.562) for <50 years; 0.599 (0.571–0.627) vs. 0.542 (0.513–0.572) for ≥50 years).

The PRS associations for women of Asian ancestry are lower compared to those of European ancestry across age groups for invasive disease (OR range: 1.62–1.64, AUC range: 0.551–0.600) and DCIS (OR range: 1.70–1.89, AUC range: 0.556–0.654). Gail model associations were weak for younger Asian-ancestry women for invasive disease and DCIS (OR range: 0.94–0.99, AUC range: 0.523–0.533), but stronger for older women (OR range: 1.82–1.88, AUC range: 0.542–0.554). Age interaction was observed only for Gail (invasive: *p* < 0.001; DCIS: *p* = 0.002).

When limiting the analysis to population-based controls only, the PRS results remained largely consistent across all disease subgroups (Appendix A). Among European-ancestry women, PRS discrimination was similar for invasive breast cancer (AUC: 0.633 [0.629–0.636] vs. 0.635 [0.632–0.638] in the full cohort) and slightly lower for DCIS (AUC: 0.623 [0.617–0.629] vs. 0.626 [0.620–0.631]). For the Gail model, discrimination in European-ancestry women decreased slightly when using population-based controls, particularly for invasive disease (AUC: 0.514 [0.510–0.517] vs. 0.492 [0.489–0.495]). In Asian-ancestry women, PRS discrimination showed modest changes: for invasive disease, the AUC decreased slightly from 0.564 [0.556–0.573] in the full cohort to 0.554 [0.542–0.565] with population-based controls, and for DCIS, from 0.587 [0.566–0.607] to 0.576 [0.555–0.598]. By contrast, Gail discrimination in Asian-ancestry women improved with population-based controls, increasing for invasive disease from 0.506 [0.497–0.514] to 0.538 [0.527–0.548] and for DCIS from 0.507 [0.486–0.528] to 0.522 [0.499–0.544]. These results suggest that PRS performance is robust to the control sampling approach, while Gail model discrimination may be more sensitive to the composition of the control group.

### 3.3. Proportions Identified as High Risk

Among women of European ancestry, Gail generally identified a greater proportion at high risk across all risk thresholds; however, in women of Asian ancestry, risk stratification was driven primarily by PRS (Appendix A). Appendix A shows the distribution of Venn diagram segments across ancestry groups, age groups, and risk thresholds. In the European-ancestry population aged ≥50 years, the minimum threshold at which PRS picked up twice as many cases as high-risk compared to controls was 1.4% (invasive) and 1.8% (DCIS). At no threshold tested did Gail identify twice as many cases as high-risk compared to controls. In younger women of European ancestry (<50 years), a risk threshold of 1% could capture twice as many invasive and DCIS cases compared to controls; for Gail, the threshold was 1.2–1.3%. At the highest threshold tested (2.5%), the proportion of invasive or DCIS cases identified as high-risk compared to controls was between 3.9 to 4.6 times. In the Asian-ancestry population aged ≥50 years, the risk threshold at which the proportion of high-risk individuals who are invasive cases is twice that of controls is approximately 2% (PRS and Gail); for DCIS, the threshold is 1.1–1.2% (PRS and Gail). For the Asian-ancestry population aged <50 years, the risk threshold at which the proportion of high-risk individuals who are invasive cases is twice that of controls is approximately 1.4% (PRS) and 2% (Gail), and for DCIS cases, the threshold is ~1.2% (PRS) and 1% (Gail).

Figure 1 shows the proportion of individuals uniquely identified by PRS, Gail, and FH across different five-year absolute-risk thresholds (0.5–2.5%). Gail uniquely identifies a large proportion of both cases (invasive and DCIS) and controls in women of European ancestry, especially among older women, at lower thresholds (<1%). At higher-risk thresholds (>~1.3%), less than 10% of the population is classified as high-risk by more than one predictor (Appendix A). However, PRS and Gail tend to have higher overlap at lower-risk thresholds. Variations in the proportions of high-risk individuals identified and overlap between predictors by country were observed (Appendix A).

#### Calibration

The empirical ROC curve does not align with the mROC curve (reference: European-ancestry, ≥50 years) for both PRS and Gail, which signals miscalibration (Figure 2). This divergence is supported by small *p*-values from the mean calibration (ranging from <2.2 × 10^−16^ to 9.00 × 10^−5^), ROC equality (from <2.2 × 10^−16^ to 0.00032), and unified calibration tests (ranging from <2.2 × 10^−16^ to 4.37 × 10^−7^) (Appendix A).

### 3.4. Drivers of Gail Model Risk

In Figure 3A, we show that in those of European ancestry, the inclusion of both FH (number of first-degree relatives with breast cancer) and prior breast biopsies yields the highest AUC values (optimal model discrimination) (AUC = 0.545 [0.540–0.551] and 0.559 [0.555–0.562] for <50 years and ≥50 years, respectively), as shown in Appendix A. For Asians < 50 years (Figure 3B), the most influential predictors are age at first live birth and number of prior biopsies (set at the reference level, as missingness is 100%) (AUC = 0.543 [0.530–0.555]). Models omitting FH performed better (Figure 3C, Appendix A). For Asian women aged ≥50 years, the best-performing model (age at first live birth + family history; AUC = 0.556 [0.544–0.568]) showed similar discrimination to the full model (AUC = 0.554 [0.543–0.566]) (Figure 3D, Appendix A).

## 4. Discussion

In this large case–control study, we analyzed the performance of PRS and Gail in 180,398 women (161,849 of European ancestry; 18,549 of Asian ancestry), stratified by age (<50 years vs. ≥50 years) and disease subtype (invasive vs. DCIS). PRS consistently outperformed traditional non-genetic risk factors (Gail model), especially in younger women and when non-genetic data were incomplete. For European-ancestry women, PRS captured inflection points where case enrichment was twice that of controls at lower absolute-risk thresholds than Gail. At the highest thresholds (2.5%), PRS enriched for 3.9–4.6× more cases than expected in both invasive and DCIS subtypes, whereas Gail failed to achieve similar discrimination. In Asian women, PRS also drove stratification, and Gail contributed minimal incremental value, particularly in younger women, where its ORs and AUCs were nearly null. PRS and Gail in groups other than the European-ancestry population aged ≥50 years require recalibration before clinical application. Driver analyses further revealed that key Gail contributors vary by ancestry: FH and prior biopsies dominate in Europeans, while reproductive factors and biopsies are most informative in younger Asians, and FH adds little incremental value. Together, these results affirm that PRS provides risk stratification by ancestry, age, and disease status, outperforming Gail across thresholds and subgroups.

PRS offers unique advantages to breast cancer risk stratification, particularly in younger women and in settings where non-genetic risk data (e.g., those used by the Gail model) are missing or unreliable (self-reporting bias, miscalibration). We observed that across age groups and disease subtypes, PRS consistently demonstrated superior discriminative accuracy compared with the Gail model. These advantages are especially pronounced in women under 50 years, for whom traditional non-genetic variables contribute minimally to risk prediction and whose risk profiles are poorly captured (and not designed to be captured) by the Gail model. Consequently, PRS identifies a large proportion of high-risk individuals missed by only considering risk factors used in routine clinical practice (age and FH) or traditional non-genetic risk factors (Gail). Given that the American College of Breast Surgeons recommends formal risk assessment beginning at age 25, integrating PRS into early risk evaluation frameworks is timely and clinically actionable [25]. Unlike traditional non-genetic risk factors, PRS enables individualized risk modeling from a younger age and supports stratified screening and prevention planning. Separating the contributions of genetic and non-genetic risk components remains relevant even when integrated models are available, especially for policy decisions in regions where PRS testing is not yet routine [26,27,28]. By quantifying the independent predictive value of PRS, policymakers can better estimate the added benefit of genetic risk stratification beyond traditional non-genetic risk factors.

These observed advantages of PRS raise the question of why genetic risk scores can provide additional predictive value beyond traditional non-genetic models. One explanation lies in the complex, multifactorial nature of cancer transformation. Cancer development can be conceptualized as a loss of regulatory control over cellular functions at both the unicellular and multicellular layers, resulting in aberrant or atavistic cell behavior [29]. This process is influenced by numerous factors, including ancestry, age, disease subtype, and other risk determinants, that may not be fully captured by clinical models like Gail. PRS, by quantifying inherited genetic susceptibility, captures part of this underlying biological risk, complementing non-genetic risk factors. The observed differences in age interactions between PRS and Gail across populations (i.e., both PRS and Gail in Europeans, but only for Gail in Asians) highlight how genetic and non-genetic contributors to risk may operate differently across populations and contexts. Therefore, integrating PRS into risk models allows for a more individualized and biologically informed assessment of breast cancer susceptibility, improving identification of high-risk individuals who may be missed by traditional risk factors alone.

Despite neither the Gail model nor PRS being explicitly developed to predict DCIS, we found that PRS stratifies DCIS risk meaningfully across age and ancestry groups, outperforming the Gail model in the ability to flag high-risk individuals [21]. In younger European-ancestry women, PRS reached the ≥2× case-to-control enrichment threshold at lower risk levels than Gail, while Gail’s enrichment was weaker and inconsistent. For older Asians, PRS again performed at least as well or better than Gail, particularly where Gail had little discriminatory power in younger women. PRS therefore adds value in DCIS risk stratification even though it was not originally designed for it. Our findings support exploring PRS as an additional component in risk models tailored for DCIS. However, the differentiation between indolent cases and those prone to progression to invasive disease will be important in the context of DCIS [30].

In European-ancestry women, the Gail model flags more individuals as “high-risk” across all risk thresholds, particularly at lower cut-points (<1%). However, PRS identifies more actual breast cancer cases. For instance, among European women aged ≥50, PRS at a 1.4% threshold for invasive disease (1.8% for DCIS) identified twice as many cases as controls, whereas the Gail model never achieves such a level of case-enrichment at any threshold. Among younger women (<50 years), PRS reaches the two-fold case vs. control ratio at a lower threshold (1.0%), compared to 1.2–1.3% for Gail. At the highest threshold (2.5%), PRS identifies 3.9–4.6× DCIS more cases than controls (i.e., it is superior to Gail at identifying women at genuinely elevated risk). Our data suggests that PRS achieves much better case–control discrimination and is able to identify high-risk individuals with fewer false positives at appropriate thresholds.

In Asian-ancestry women, risk stratification is driven primarily by PRS. Both PRS and Gail reach the two-fold enrichment threshold at similar absolute risk levels (~1.4–2.0% for invasive disease, ~1.1–1.2% for DCIS), but Gail contributes minimally in younger Asian women (where its ORs and discrimination are particularly weak). The unique Venn diagram segmentation further highlights that PRS uniquely flags high-risk individuals who are missed by Gail or FH. In sum, these results demonstrate that PRS adds value, especially in populations or age groups where non-genetic predictors are weak. However, it is important to weigh that benefit against expected increases in false positives and overdiagnoses [31].

While both models show reasonable discrimination, they fail to assign accurate absolute risks, which can potentially over- or underestimate risks and mislead clinical decisions. Miscalibration is a common issue when applying PRS or Gail model estimates derived from European-ancestry datasets to independent samples with different case mix characteristics [32,33,34]. In practice, a well-calibrated model typically has a calibration slope close to 1 and an intercept near 0. In the absence of a universally accepted numerical threshold for deviations, it is important to consider miscalibration as clinically meaningful if it leads to over- or underestimation of risk that could influence clinical decision-making [35]. Therefore, in real-world applications, recalibration or adjustment protocols, such as updating baseline incidence rate or performing logistic recalibration, will be necessary to ensure accurate absolute-risk predictions before clinical implementation.

We showed that the Gail model’s performance varies across populations due to differences in risk factor distributions. These variations highlight the need for adapting risk models to specific population characteristics. While our findings may not generalize beyond European and Asian populations, prior work has shown that PRSs derived from European GWAS can retain predictive potential within each ancestry group, highlighting the importance of evaluating PRS performance across diverse populations [36]. Additionally, challenges in accurately completing clinical fields have limited the widespread use of the Gail model in the general population [37].

Our study benefits from a large, multi-ancestry case–control cohort of 180,398 women (161,849 of European ancestry; 18,549 of Asian ancestry), providing substantial power to compare PRS versus Gail model performance across clinically important subgroups of age (<50 years vs. ≥50 years) and disease subtype (invasive vs. DCIS). Where the previous literature typically used one threshold for risk cut-off, we quantified high-risk enrichment across absolute-risk thresholds. We also performed driver analyses, revealing that key contributors to Gail’s performance differ by population.

However, our study has limitations. The heterogeneity in data sources and cohort methods may introduce variability in risk-factor measurement and disease ascertainment. The high proportion of missing data for key Gail model variables represents an important limitation, particularly among European participants. Sensitivity analyses indicated that the model’s discrimination for invasive breast cancer was more affected by assumptions about missing data than for DCIS, likely due to the heavy weighting of family history and biopsy variables in the Gail model. The larger changes in AUC observed among Europeans suggest that missingness in these predictors contributed to greater uncertainty in model performance. Additionally, while we showed PRS superiority for case enrichment, formal calibration assessments and clinical thresholds were not exhaustively validated in every subgroup. Asian-specific polygenic risk scores could improve breast cancer risk prediction and risk stratification. However, they were not evaluated in our study. Finally, our models did not include other potentially informative predictors, such as mammographic density, lifestyle factors, or hormone use, that may further refine individualized risk.

## 5. Conclusions

Our results highlight ancestry- and age-specific performance of PRS and Gail model across risk thresholds and strengthen the case for incorporating PRS into breast cancer risk stratification. PRS adds value risk stratification beyond traditional tools, especially in younger women and Asian-ancestry populations.

## Figures and Tables

**Figure 1 cancers-17-03561-f001:**
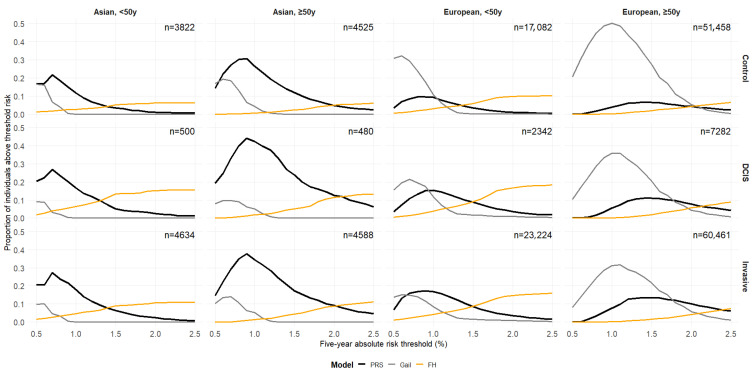
Proportions of individuals uniquely identified as high-risk by the polygenic risk score (PRS), the Gail model, and family history (FH) at varying absolute-risk thresholds for PRS and Gail. FH is binary (presence vs. absence of a first-degree relative with breast cancer), so its classification as high-risk does not have a varying threshold. Each line represents one identification method (black: PRS only; grey: Gail only; orange: FH only). These values match the non-overlapping regions (unique segments) if Venn diagrams were to be plotted for each threshold. y: years.

**Figure 2 cancers-17-03561-f002:**
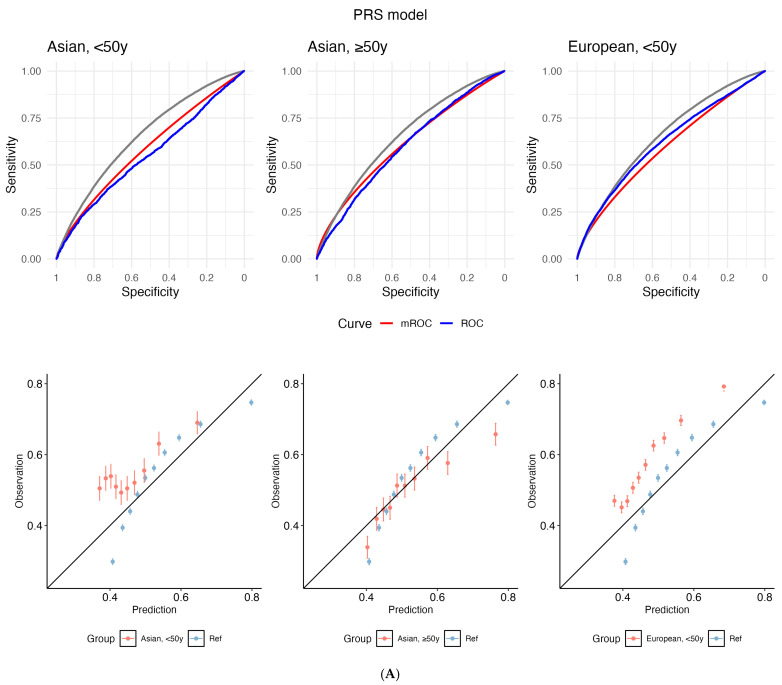
Receiver operating characteristic (ROC) curves (top panel) and calibration plots (bottom) for (**A**) polygenic risk score (PRS) and (**B**) the Gail model. The grey line depicts the empirical ROC of the reference population: European, ≥50 years). The model-based ROC (mROC) curve (red) depicts the ROC performance expected if the prediction model is perfectly calibrated in the external population (i.e., European, <50 years; Asian, ≥50 years; Asian, <50 years). The empirical ROC curve (blue) shows discrimination based on observed outcomes. Alignment between red and blue curves implies good calibration; divergence suggests miscalibration after accounting for case mix. Calibration plots are used to assess the agreement between predictions and observations in different percentiles of the predicted values. The black diagonal line represents ideal calibration, where the predicted probabilities match the observed frequencies. Deviations from this diagonal line indicate either overconfidence or underconfidence in the model’s predictions. y: years.

**Figure 3 cancers-17-03561-f003:**
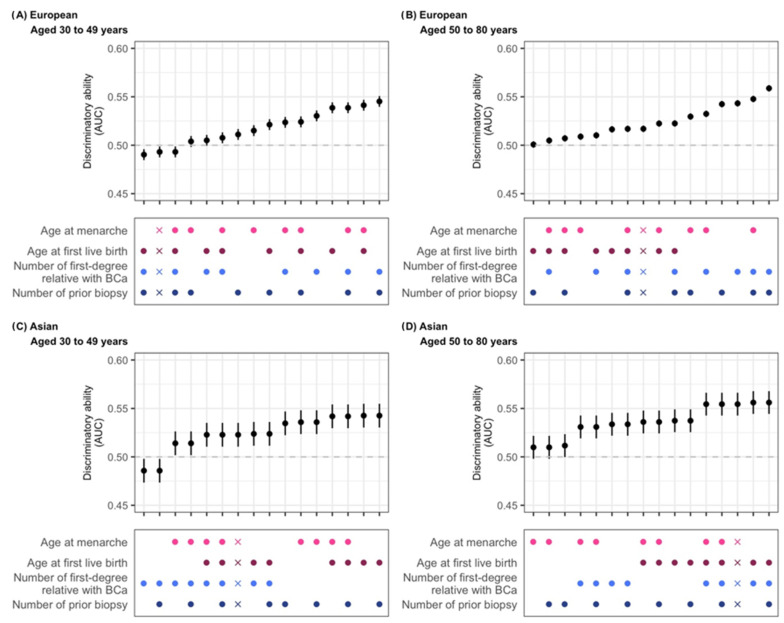
Discriminatory ability of risk factor combinations in the Gail model. Five-year absolute risk was calculated using the R package “BCRA” and used to predict the risk of invasive breast cancer in a case–control study. Grey dashed line: AUC=0.5, model performs no better than random chance. Dots represent risk factors included in the model (factors not considered are set to missing), and X indicates the model with all risk factors with the addition of atypical hyperplasia. European: women of European ancestry; Asian: women of Asian ancestry. As data on prior biopsies were unavailable for Asian-ancestry participants, retaining this variable in the model produced the same effect as assuming all participants of Asian ancestry had missing values for this variable (i.e., lowest-risk group assigned by default).

**Table 1 cancers-17-03561-t001:** Characteristics of 161,849 European-ancestry (European) and 18,549 Asian-ancestry (Asian) individuals between the ages of 30 and 80 years. IQR: interquartile range; Family history: number of first-degree relatives with breast cancer; Invasive: invasive breast cancer; DCIS: ductal carcinoma in situ.

	European, n = 161,849			Asian, n = 18,549		
	Control n = 68,540 (42%)	Invasive n = 83,685 (52%)	DCIS n = 9624 (6%)	Control n = 8347 (45%)	Invasive n = 9222 (50%)	DCIS n = 980 (5%)
**Median age (at interview/diagnosis), years (IQR)**	57 (50 to 64)	57 (49 to 65)	55 (50 to 63)	50 (44 to 58)	49 (43 to 57)	49 (43 to 56)
**Age at menarche, n (%)**						
<12 years	9825 (14)	10,859 (13)	1733 (18)	508 (6)	586 (6)	61 (6)
12 to 14 years	30,243 (44)	33,476 (40)	4546 (47)	3080 (37)	3278 (36)	326 (33)
≥14 years	21,133 (31)	23,169 (28)	2758 (29)	4321 (52)	4186 (45)	467 (48)
Unknown	7339 (11)	16,181 (19)	587 (6)	438 (5)	1172 (13)	126 (13)
**Age at first full-term pregnancy, n (%)**						
Nulliparous	8319 (12)	10,769 (13)	1520 (16)	1005 (12)	1236 (13)	158 (16)
<20 years	5427 (8)	6138 (7)	794 (8)	316 (4)	362 (4)	29 (3)
20 to 24 years	21,863 (32)	21,927 (26)	2899 (30)	1948 (23)	1810 (20)	135 (14)
25 to 29 years	17,543 (26)	18,174 (22)	2382 (25)	2836 (34)	3030 (33)	293 (30)
≥30 years	8487 (12)	9973 (12)	1386 (14)	1149 (14)	1554 (17)	156 (16)
Unknown	6901 (10)	16,704 (20)	643 (7)	1093 (13)	1230 (13)	209 (21)
**Family history, n (%)**						
No	45,629 (67)	48,903 (58)	4005 (42)	7181 (86)	7500 (81)	708 (72)
1	5348 (8)	10,256 (12)	1115 (12)	437 (5)	876 (9)	114 (12)
≥2	791 (1)	2064 (2)	305 (3)	63 (1)	93 (1)	13 (1)
Unknown	16,772 (24)	22,462 (27)	4199 (44)	666 (8)	753 (8)	145 (15)
**Number of breast biopsy, n (%)**						
No	930 (1)	3181 (4)	96 (1)	0 (0)	0 (0)	0 (0)
1	277 (0)	3148 (4)	216 (2)	0 (0)	0 (0)	0 (0)
≥2	103 (0)	1822 (2)	148 (2)	0 (0)	0 (0)	0 (0)
Unknown	67,230 (98)	75,534 (90)	9164 (95)	8347 (100)	9222 (100)	980 (100)
**Atypical hyperplasia, n (%)**						
No	930 (1)	3181 (4)	96 (1)	0 (0)	0 (0)	0 (0)
Yes	5 (0)	49 (0)	8 (0)	0 (0)	0 (0)	0 (0)
Unknown	67,605 (99)	80,455 (96)	9520 (99)	8347 (100)	9222 (100)	980 (100)
**Median five-year absolute risk by Gail** **(IQR)**	1.25 (0.92 to 1.64)	1.25 (0.89 to 1.73)	1.29 (0.98 to 1.71)	0.61 (0.45 to 0.78)	0.61 (0.44 to 0.80)	0.61 (0.38 to 0.81)
**Protein truncating variants (9 Genes)**						
No	24,215 (35)	27,926 (33)	1662 (17)	1091 (13)	2049 (22)	317 (32)
Yes	583 (1)	1927 (2)	74 (1)	24 (0)	129 (1)	7 (1)
Unknown	43,742 (64)	53,832 (64)	7888 (82)	7232 (87)	7044 (76)	656 (67)
**Polygenic risk score (PRS)**	−0.45 (−0.86 to −0.04)	−0.09 (−0.51 to 0.32)	−0.15 (−0.55 to 0.27)	0.16 (−0.20 to 0.53)	0.37 (−0.01 to 0.76)	0.45 (0.05 to 0.83)
**Median five-year absolute risk by PRS (IQR)**	0.69 (0.46 to 1.05)	0.95 (0.62 to 1.44)	0.91 (0.61 to 1.37)	0.62 (0.40 to 0.95)	0.74 (0.44 to 1.15)	0.79 (0.44 to 1.28)

**Table 2 cancers-17-03561-t002:** Association between five-year absolute breast cancer risk (modeled as a continuous variable, estimated using the polygenic risk score (PRS) versus the Gail model) and the development of invasive breast cancer or ductal carcinoma in situ (DCIS), stratified by ancestry and age groups. OR: odds ratios; CI: confidence intervals; AUC: area under the curve. ORs and AUCs are estimated separately in age-stratified logistic regression models (<50 years and ≥50 years). *p*-values for interaction (risk score × Age) were obtained from separate logistic regression models that included the risk score, age (treated as a continuous variable), and their interaction term; the interaction test assessed statistical heterogeneity of the risk-score effect by age.

	All Ages		<50 Years		≥50 Years	
	OR (95% CI)	AUC (95% CI)	OR (95% CI)	AUC (95% CI)	OR (95% CI)	AUC (95% CI)
**Invasive**						
*European, n = 152,225*						
PRS	1.97 (1.94 to 2.01)	0.635 (0.632 to 0.638)	2.51 (2.39 to 2.62)	0.622 (0.617 to 0.628)	2.06 (2.02 to 2.11)	0.653 (0.650 to 0.656)
P interaction (PRS × Age)	<0.001					
Gail	1.12 (1.11 to 1.14)	0.492 (0.489 to 0.495)	1.35 (1.29 to 1.40)	0.493 (0.487 to 0.499)	1.18 (1.16 to 1.19)	0.517 (0.514 to 0.520)
P interaction (Gail × Age)	<0.001					
PRS and Gail combined		0.635 (0.632 to 0.638)		0.621 (0.616 to 0.627)		0.654 (0.651 to 0.658)
PRS	1.97 (1.93 to 2.00)		2.46 (2.35 to 2.58)		2.04 (2.00 to 2.08)	
Gail	1.01 (1.00 to 1.03)		1.06 (1.02 to 1.10)		1.13 (1.11 to 1.15)	
*Asian, n = 17,569*						
PRS	1.48 (1.41 to 1.56)	0.564 (0.556 to 0.573)	1.62 (1.47 to 1.78)	0.551 (0.539 to 0.563)	1.64 (1.53 to 1.75)	0.600 (0.588 to 0.611)
P interaction (PRS × Age)	0.833					
Gail	1.19 (1.09 to 1.30)	0.506 (0.497 to 0.514)	0.94 (0.81 to 1.08)	0.523 (0.511 to 0.535)	1.82 (1.61 to 2.07)	0.554 (0.543 to 0.566)
P interaction (Gail × Age)	<0.001					
PRS and Gail combined		0.564 (0.556 to 0.573)		0.566 (0.554 to 0.578)		0.611 (0.599 to 0.622)
PRS	1.48 (1.40 to 1.56)		1.76 (1.58 to 1.95)		1.61 (1.51 to 1.73)	
Gail	1.00 (0.91 to 1.09)		0.69 (0.59 to 0.81)		1.75 (1.54 to 1.99)	
**DCIS**						
*European, n = 78,164*						
PRS	1.63 (1.59 to 1.68)	0.626 (0.620 to 0.631)	2.56 (2.37 to 2.78)	0.657 (0.645 to 0.669)	1.56 (1.51 to 1.61)	0.620 (0.613 to 0.626)
P interaction (PRS × Age)	<0.001					
Gail	1.23 (1.20 to 1.26)	0.537 (0.531 to 0.543)	2.28 (2.10 to 2.49)	0.610 (0.597 to 0.622)	1.19 (1.16 to 1.22)	0.519 (0.512 to 0.526)
P interaction (Gail × Age)	<0.001					
PRS and Gail combined		0.622 (0.616 to 0.628)		0.669 (0.657 to 0.681)		0.618 (0.611 to 0.624)
PRS	1.59 (1.55 to 1.64)		2.29 (2.11 to 2.48)		1.54 (1.50 to 1.59)	
Gail	1.15 (1.12 to 1.18)		1.92 (1.76 to 2.10)		1.15 (1.12 to 1.19)	
*Asian, n = 9327*						
PRS	1.67 (1.52 to 1.83)	0.587 (0.566 to 0.607)	1.70 (1.42 to 2.03)	0.556 (0.528 to 0.584)	1.89 (1.69 to 2.12)	0.654 (0.628 to 0.680)
P interaction (PRS × Age)	0.313					
Gail	1.25 (1.03 to 1.52)	0.507 (0.486 to 0.528)	0.99 (0.71 to 1.38)	0.533 (0.505 to 0.562)	1.88 (1.46 to 2.41)	0.542 (0.513 to 0.572)
P interaction (Gail × Age)	0.002					
PRS and Gail combined		0.587 (0.566 to 0.607)		0.565 (0.537 to 0.593)		0.665 (0.640 to 0.691)
PRS	1.66 (1.52 to 1.83)		1.78 (1.48 to 2.15)		1.88 (1.68 to 2.11)	
Gail	1.01 (0.82 to 1.25)		0.72 (0.50 to 1.05)		1.80 (1.39 to 2.32)	

For Gail, in younger European-ancestry individuals with invasive breast cancer, the OR was 1.35 (1.29–1.40, AUC: 0.493 [0.487–0.499]). Among older individuals with invasive disease, the OR was 1.18 (1.16–1.19, AUC: 0.517 [0.514–0.520]). For DCIS, the younger group had a markedly higher OR of 2.28 (2.10–2.49, AUC: 0.610 [0.597–0.622]). In contrast, older DCIS cases showed an OR of 1.19 (1.16–1.22, AUC: 0.519 [0.512–0.526]). Age interactions were found for both invasive breast cancer (PRS, *p* < 0.001; Gail, *p* < 0.001) and DCIS (PRS, *p* < 0.001; Gail, *p* < 0.001).

## Data Availability

The data used in our analyses are available upon reasonable request through BCAC, subject to data access committee approval.

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
