# Peer review of "Threshold-Based Overlap of Breast Cancer High-Risk Classification Using Family History, Polygenic Risk Scores, and Traditional Risk Models in 180,398 Women"

_cancers, 2025, doi:10.3390/cancers17213561_

Round 1
Reviewer 1 Report
Comments and Suggestions for Authors
The paper compares polygenic risk scores (PRS), traditional Gail model, and family histiry (FH) in terms of breast cancer risk estimation.
The authors investigate how the overlap varies by ancestry, age, disease status (invasive, DCIS, and controls) and risk threshold, and conclude that PRS adds value to risk stratification beyond traditional tools, especially in younger women and non-European ancestry populations.
There are various unclear descriptions and presentation that need be improved for publication.
Major comments
Abstract
"Significant age interactions": The "significance" should be avoided. Please rephrase it by using p-value.
"OR" and "AUC" are undefined.
"especially in younger women and non-European ancestry populations": The authors did not compare non-European ancestry populations other than Asians.
"Calibration analysis revealed misalignment between observed and predicted risks": Please detail what the sentence concerns.
Page 8
"The 313-variant breast cancer PRS" is undefined. Please also provide the detail of the variants including allele frequencies stratified by ancestry, age, and disease status.
"Age‑based effect modification is tested by including an age×exposure (PRS or Gail) interaction term": Please mention what model is used to consider the interaction term. Is this the logistic regression model?
Page 9
"Significantly" is undefined.
Page 11
"OR" is undefined. Please describe how the OR (odds ratio) is estimated. Is this from the logistic regression model?
Please add the confidence interval for AUC in the main text as done in Table 2.
"The interaction with age was significant": "Significant" is undefined.
Page 12
"Significant" is undefined.
Page 13
"highly significant p-values" is undefined.
Page 15
"In Asians ≥50y, model performance was not significantly different": "significantly different" is undefined. Please also add the p-value.
Figure 1
"Proportion of individuals uniquely identified as high-risk by the polygenic risk score (PRS), the Gail model, and family history (FH) at varying absolute risk thresholds for PRS and Gail": I don't know why the authors do not mention the threshold for FH.
This figure is difficult to read as it shows the proportion of individuals uniquely identified and does not show the overlap between the models. As the title of the paper emphasize the overlap, it is better to use a better plot emphasizing the overlap rather than the proportion of individuals uniquely identified.
It is more informative to show the overlap between methods by varying the threshold for understanding which models are overlapped.
Page 17
"Our findings may not generalize beyond European and Asian populations": There is a study regarding generalizability of PRS to different populations, Fritsche et al. (2021). It is worth discussing.
Reference:
Fritsche LG, Ma Y, Zhang D, Salvatore M, Lee S, Zhou X, et al. (2021) On cross-ancestry cancer polygenic risk scores. PLoS Genet 17(9): e1009670. https://doi.org/10.1371/journal.pgen.1009670
Author Response
The paper compares polygenic risk scores (PRS), traditional Gail model, and family histiry (FH) in terms of breast cancer risk estimation.
The authors investigate how the overlap varies by ancestry, age, disease status (invasive, DCIS, and controls) and risk threshold, and conclude that PRS adds value to risk stratification beyond traditional tools, especially in younger women and non-European ancestry populations.
There are various unclear descriptions and presentation that need be improved for publication.
Major comments
Abstract
"Significant age interactions": The "significance" should be avoided. Please rephrase it by using p-value.
Our response:
We have edited the result to:
“Age interactions were observed for both PRS (p<0.001) and Gail (p<0.001) in Europeans, whereas in Asians, age interaction was observed only for Gail (invasive: p<0.001; DCIS: p=0.002).”
"OR" and "AUC" are undefined.
Our response:
Added:
“Odds ratios (ORs) from logistic regression models and area under the receiver operating characteristic curve (AUC) were estimated.”
"especially in younger women and non-European ancestry populations": The authors did not compare non-European ancestry populations other than Asians.
Our response:
Edited:
“PRS adds value to risk stratification beyond traditional tools, especially in younger women and Asian-ancestry populations.”
"Calibration analysis revealed misalignment between observed and predicted risks": Please detail what the sentence concerns.
Our response:
Edited:
“Calibration analysis, comparing empirical and model-based ROC curves, showed divergence for both PRS and Gail (p<0.001), which indicates miscalibration.”
Page 8
"The 313-variant breast cancer PRS" is undefined. Please also provide the detail of the variants including allele frequencies stratified by ancestry, age, and disease status.
Our response:
We have added:
“Among the multiple PRSs available for breast cancer, we used the 313-variant breast cancer PRS developed by the Breast Cancer Association Consortium (BCAC). This PRS was selected because it has been extensively validated in large-scale studies across diverse populations and has demonstrated strong and consistent associations with breast cancer risk and reproducible discriminatory performance (14)(20). Details of the variants including allele frequencies stratified by ancestry, age, and disease status in our analytical dataset are presented in Supplementary Table 2.”
"Age‑based effect modification is tested by including an age×exposure (PRS or Gail) interaction term": Please mention what model is used to consider the interaction term. Is this the logistic regression model?
Our response:
Edited in Abstract:
“Odds ratios (ORs) from logistic regression models and area under the receiver operating characteristic curve (AUC) were estimated.”
In main text:
“Age‑based effect modification is tested by including an age × exposure (PRS or Gail) interaction term in logistic regression models.”
Page 9
"Significantly" is undefined.
Our response:
We have removed “significantly” from the summary sentence and the following sentences with explicit pvalues:
“Excluded Europeans differed from those included regarding age at first full-term pregnancy, and missing data on age at menarche was more common among excluded Europeans. Across all subgroups, excluded individuals had lower 5-year absolute risk according to Gail (p<0.001). PRS sum score was higher in excluded Europeans (-0.237 vs. -0.247; p=0.002) and in excluded Asians (0.308 vs. 0.279; p=0.002). For Asians, excluded individuals also had higher PRS-based 5-year absolute risk (0.716 vs. 0.681; p<0.001).”
Page 11
"OR" is undefined. Please describe how the OR (odds ratio) is estimated. Is this from the logistic regression model?
Our response:
Now defined in Methods:
“We assessed the relationship between estimated five-year absolute breast cancer risk (modeled as a continuous variable using PRS and the Gail model) and invasive breast cancer or DCIS. Logistic regression models were fitted to estimate odds ratios (ORs) and corresponding 95% confidence intervals (CIs). Analyses were stratified by disease (invasive, DCIS), genetic ancestry (Asian, European) and age group (<50y, ≥50y). Age‑based effect modification is tested by including an age × exposure (PRS or Gail) interaction term in logistic regression models.”
Please add the confidence interval for AUC in the main text as done in Table 2.
Our response:
Revised to include CIs:
“Performance of risk models
In invasive breast cancer among individuals of European ancestry, younger women (<50y) exhibited a stronger PRS association (OR:2.51 [2.39-2.62]) but lower discrimination (AUC:0.622 [0.617-0.628]), whereas in older women (≥50y), the PRS effect was weaker (OR:2.06 [2.02-2.11]) with higher discrimination (AUC:0.653 [0.650-0.656]) (Table 2). In contrast, for DCIS, younger women showed both stronger association and better discrimination (OR:2.56 [2.37-2.78]; AUC=0.657 [0.645-0.669]), while older women had lower OR and AUC (OR:1.56 [1.51-1.61]; AUC=0.620 [0.613-0.626]).
For Gail, in younger European-ancestry individuals with invasive breast cancer, the OR was 1.35 (1.29-1.40, AUC:0.493 [0.487-0.499]) Among older individuals with invasive disease, the OR was 1.18 (1.16-1.19, AUC:0.517 [0.514-0.520]). For DCIS, the younger group had a markedly higher OR of 2.28 (2.10-2.49, AUC: 0.610 [0.597-0.622]) In contrast, older DCIS cases showed an OR of 1.19 (1.16-1.22, AUC:0.519 [0.512-0.526]). Age interactions were found for both invasive breast cancer (PRS, p<0.001; Gail, p<0.001) and DCIS (PRS, p<0.001; Gail, p<0.001).
The PRS associations for women of Asian ancestry are lower compared to those of European ancestry across ages groups for invasive disease (OR range: 1.62-1.64, AUC range: 0.551-0.600) and DCIS (OR range: 1.70-1.89, AUC range: 0.556-0.654). Gail model-associations were weak for younger Asian-ancestry women for invasive disease and DCIS (OR range: 0.94-0.99, AUC range: 0.523-0.533), but stronger for older women (OR range: 1.82-1.88, AUC range: 0.542-0.554). Age interaction was observed only for Gail (invasive: p<0.001; DCIS: p=0.002).
When limiting the analysis to population-based controls only, the PRS results remained largely consistent across all disease subgroups (Supplementary Table 11). Among European-ancestry women, PRS discrimination was similar for invasive breast cancer (AUC:0.633 [0.629-0.636] vs. 0.635 [0.632-0.638] in the full cohort) and slightly lower for DCIS (AUC:0.623 [0.617-0.629] vs. 0.626 [0.620-0.631]). For the Gail model, discrimination in European women decreased slightly when using population-based controls, particularly for invasive disease (AUC:0.514 [0.510-0.517] vs. 0.492 [0.489-0.495]). In Asian-ancestry women, PRS discrimination showed modest changes: for invasive disease, the AUC decreased slightly from 0.564 [0.556-0.573] in the full cohort to 0.554 [0.542-0.565] with population-based controls, and for DCIS, from 0.587 [0.566-0.607] to 0.576 [0.555-0.598]. By contrast, Gail discrimination in Asian women improved with population-based controls, increasing for invasive disease from 0.506 [0.497-0.514] to 0.538 [0.527-0.548] and for DCIS from 0.507 [0.486-0.528] to 0.522 [0.499-0.544]. These results suggest that PRS performance is robust to the control sampling approach, while Gail model discrimination may be more sensitive to the composition of the control group.”
"The interaction with age was significant": "Significant" is undefined.
Our response:
Edited:
“Age interactions were found for both invasive breast cancer (PRS, p<0.001; Gail, p<0.001) and DCIS (PRS, p<0.001; Gail, p<0.001).”
Page 12
"Significant" is undefined.
Our response:
Revised sentence:
“Age interaction was observed only for Gail (invasive: p<0.001; DCIS: p=0.002).”
Page 13
"highly significant p-values" is undefined.
Our response:
Revised sentence:
“The divergence is supported by small p-values from the mean calibration (A), mROC equality (ranging from <2.2×10⁻¹⁶ to 9.00×10⁻5) (B) (ranging from <2.2×10⁻¹⁶ to 0.00032), and unified (U) tests (ranging from <2.2×10⁻¹⁶ to 4.37×10⁻⁷) (Supplementary Table 13).”
Page 15
"In Asians ≥50y, model performance was not significantly different": "significantly different" is undefined. Please also add the p-value.
Our response:
We have added Supplementary Table 14 which shows the AUC and 95% CI corresponding to Figure 3. We have revised the text to replace “not significantly different” with “comparable,” as no formal statistical test was performed.
“In Figure 3A and Figure 3B, we show that in Europeans, the inclusion of both FH (number of first-degree relatives with breast cancer) and prior breast biopsies yields the highest AUC values (optimal model discrimination) (AUC=0.545 [0.540-0.551] and 0.559 [0.555-0.562] for <50y and ≥50y, respectively) Supplementary Table 14). For Asians <50y, the most influential predictors are age at first live birth and number of prior biopsies (set at reference level, as missingness is 100%) (AUC=0.543 [0.530-0.555]). Models omitting FH performed better. (Figure 3C, Supplementary Table 14). For Asian women aged ≥50 years, the best-performing model (age at first live birth + family history; AUC=0.556 [0.544-0.568]) showed similar discrimination to the full model (AUC=0.554 [0.543-0.566] (Figure 3D, Supplementary Table 14).”
Figure 1
"Proportion of individuals uniquely identified as high-risk by the polygenic risk score (PRS), the Gail model, and family history (FH) at varying absolute risk thresholds for PRS and Gail": I don't know why the authors do not mention the threshold for FH.
Our response:
FH is a binary variable (presence vs. absence of first-degree relatives with breast cancer), unlike PRS and Gail, which are continuous and allow risk stratification at varying absolute risk thresholds. Therefore, FH does not have a range of thresholds, and its “high-risk” classification is binary by definition. We have clarified this point in the figure legend:
“Figure 1 Proportion of individuals uniquely identified as high-risk by the polygenic risk score (PRS), the Gail model, and family history (FH) at varying absolute risk thresholds for PRS and Gail. FH is binary (presence vs. absence of a first-degree relative with breast cancer), so its classification as high-risk does not have a varying threshold. Each line represents one identification method (black: PRS only; grey: Gail only; orange: FH only). These values match the non-overlapping regions (unique segments) if Venn diagrams were to be plotted for each threshold.”
This figure is difficult to read as it shows the proportion of individuals uniquely identified and does not show the overlap between the models. As the title of the paper emphasize the overlap, it is better to use a better plot emphasizing the overlap rather than the proportion of individuals uniquely identified.
It is more informative to show the overlap between methods by varying the threshold for understanding which models are overlapped.
Our response:
We thank the reviewer for this suggestion. We agree that visualizing the overlap can be informative. To address this, Supplementary Figure 5 shows the proportion of individuals identified as high-risk by multiple factors (PRS, Gail, and FH) across varying absolute-risk thresholds. These values correspond directly to the overlapping regions in Venn diagrams for each threshold. We believe this provides a clear representation of both unique and shared high-risk classifications, complementing Figure 1 while preserving readability.
Supplementary Figure 5. Proportion of individuals identified as high-risk by multiple factors across varying absolute-risk thresholds. PRS: polygenic risk score; Gail: the Gail model; FH: family history. These values match the overlapping regions from the corresponding Venn diagrams for each threshold.
Page 17
"Our findings may not generalize beyond European and Asian populations": There is a study regarding generalizability of PRS to different populations, Fritsche et al. (2021). It is worth discussing.
Reference:
Fritsche LG, Ma Y, Zhang D, Salvatore M, Lee S, Zhou X, et al. (2021) On cross-ancestry cancer polygenic risk scores. PLoS Genet 17(9): e1009670. https://doi.org/10.1371/journal.pgen.1009670
Our response:
Added:
“We showed that the Gail model's performance varies across populations due to differences in risk factor distributions. These variations highlight the need for adapting risk models to specific population characteristics. While our findings may not generalize beyond European and Asian populations, prior work has shown that PRSs derived from European GWAS can retain predictive potential within each ancestry group, highlighting the importance of evaluating PRS performance across diverse populations (34).”
Reviewer 2 Report
Comments and Suggestions for Authors
Review of the article "Threshold‑based overlap of breast cancer high‑risk classification using family history, polygenic risk scores and traditional risk models in 180,398 women
In my opinion, the article is interesting and valuable.
In my opinion, the Authors should try to add (for example, in the Discussion section) the interpretations of possible reasons why "PRS adds value to risk stratification beyond traditional tools". In accordance with the article (https://doi.org/10.3390/ijms23074017): "cancer transformation (i.e., a change in normal cell-fate to cancerous/atavistic cell fate) occurs as a result of loss of control over functionalities of the unicellular layer, resulting in a loss of control over atavistic functionalities" and "Cancer transformation can occur as a result of huge disturbances in functionalities of the multicellular layer that normally control activity of atavistic functionalities" indicating that there can be a lot of factors that can affect cancer transformation (i.e., there can be a lot of factors that can affect this loss of control), among others, ancestry (i.e., in your article, European-ancestry and Asian-ancestry), age, disease/cancer type, risk threshold. For this reason, PRS and Gail can give different interactions, which is in accordance with "Significant age interactions were observed for PRS and Gail in Europeans, but only for Gail in Asians" from Abstract in your article. In accordance with your article, it is visible that PRS in some cases can give better predictions, better taking into account factors that can affect this loss of control. I propose the Authors extend this interpretation and present it in the article, which should increase the value and attractiveness of the article.
Author Response
Review of the article "Threshold‑based overlap of breast cancer high‑risk classification using family history, polygenic risk scores and traditional risk models in 180,398 women
In my opinion, the article is interesting and valuable.
In my opinion, the Authors should try to add (for example, in the Discussion section) the interpretations of possible reasons why "PRS adds value to risk stratification beyond traditional tools". In accordance with the article (https://doi.org/10.3390/ijms23074017): "cancer transformation (i.e., a change in normal cell-fate to cancerous/atavistic cell fate) occurs as a result of loss of control over functionalities of the unicellular layer, resulting in a loss of control over atavistic functionalities" and "Cancer transformation can occur as a result of huge disturbances in functionalities of the multicellular layer that normally control activity of atavistic functionalities" indicating that there can be a lot of factors that can affect cancer transformation (i.e., there can be a lot of factors that can affect this loss of control), among others, ancestry (i.e., in your article, European-ancestry and Asian-ancestry), age, disease/cancer type, risk threshold. For this reason, PRS and Gail can give different interactions, which is in accordance with "Significant age interactions were observed for PRS and Gail in Europeans, but only for Gail in Asians" from Abstract in your article. In accordance with your article, it is visible that PRS in some cases can give better predictions, better taking into account factors that can affect this loss of control. I propose the Authors extend this interpretation and present it in the article, which should increase the value and attractiveness of the article.
Our response:
Added:
“These observed advantages of PRS raise the question of why genetic risk scores can provide additional predictive value beyond traditional non-genetic models. One explanation lies in the complex, multifactorial nature of cancer transformation. Cancer development can be conceptualized as a loss of regulatory control over cellular functions at both the unicellular and multicellular layers, resulting in aberrant or atavistic cell behavior (28). This process is influenced by numerous factors, including ancestry, age, disease subtype, and other risk determinants, that may not be fully captured by clinical models like Gail. PRS, by quantifying inherited genetic susceptibility, captures part of this underlying biological risk, complementing non-genetic risk factors. The observed differences in age interactions between PRS and Gail across populations (i.e. both PRS and Gail in Europeans, but only for Gail in Asians) highlight how genetic and non-genetic contributors to risk may operate differently across populations and contexts. Therefore, integrating PRS into risk models allows for a more individualized and biologically informed assessment of breast cancer susceptibility, improving identification of high-risk individuals who may be missed by traditional risk factors alone.”
Reviewer 3 Report
Comments and Suggestions for Authors
The manuscript presents a case–control study comparing the performance of polygenic risk score (PRS) and a non-genetic breast cancer risk predictor (Gail model) in identifying high-risk individuals across ancestry, age, and disease subtypes. The study addresses an important topic, and the results may provide useful insights. Below are some comments and suggestions for further improvement:
Major Comments
- The comparison of PRS and Gail was conducted using the odds ratio (OR) estimates of the interaction term between the two risk scores and age. However, the interaction term reflects heterogeneity of the 5-year risk estimates across age groups. To more appropriately capture the effect of PRS and Gail, both the main effect (PRS/Gail term) and the interaction term should be included, with the results described at specific ages if the interaction is significant. The current OR estimates alone may not adequately represent the effects.
- The Gail model requires the number of prior breast biopsies for calculation. As most participants in the study population have missing values for this variable, clarification is needed on how this was handled. Could this missingness have influenced the performance of the Gail model? An additional explanation would be helpful.
- Potential drivers of Gail risk could be explored by including each factor in the logistic regression model. The methods section does not provide information on missing data handling. It may be useful to discuss possible imputation methods, particularly for missing family history variables.
- For statistical comparisons, please consider the use of Fisher’s exact test where appropriate.
- To more directly compare the effects of PRS and Gail, it may be informative to include both indices simultaneously in the regression model, as they have been studied as independent risk factors.
Minor Comments
- The function “pnorm” referenced in the manuscript is an R function, not a package. Please revise accordingly.
- The manuscript would benefit from additional editorial revision to improve readability and clarity.
Author Response
The manuscript presents a case–control study comparing the performance of polygenic risk score (PRS) and a non-genetic breast cancer risk predictor (Gail model) in identifying high-risk individuals across ancestry, age, and disease subtypes. The study addresses an important topic, and the results may provide useful insights. Below are some comments and suggestions for further improvement:
Major Comments
- The comparison of PRS and Gail was conducted using the odds ratio (OR) estimates of the interaction term between the two risk scores and age. However, the interaction term reflects heterogeneity of the 5-year risk estimates across age groups. To more appropriately capture the effect of PRS and Gail, both the main effect (PRS/Gail term) and the interaction term should be included, with the results described at specific ages if the interaction is significant. The current OR estimates alone may not adequately represent the effects.
Our response:
We thank the reviewer for this thoughtful comment. In our analysis, age-stratified logistic regression models were fitted separately for women aged <50 and ≥50 years to provide directly interpretable, age-specific associations for each risk score. This approach allows the estimation of the effect of PRS and Gail within each age group without relying on model-based interaction effects, which can be challenging to interpret clinically.
The corresponding interaction between age (modeled as a continuous variable) and each risk score was evaluated in separate logistic regression models, and P interaction values are reported to indicate whether heterogeneity by age was statistically significant. Because our goal was to describe risk score performance at clinically relevant age groups rather than to model the continuous age-by-score relationship, we retained the stratified results as the primary presentation. We believe this approach conveys the findings more transparently and is consistent with common practice in risk model evaluation studies.
We have clarified in Methods:
“This approach provides directly interpretable, age-specific effect estimates without relying on interaction terms. Formal interaction tests between age (modeled as a continuous variable) and each risk score were conducted separately and reported as P interaction values to assess potential heterogeneity of effects by age.”
In Table 2 and Supplementary Table 11:
“ORs and AUCs are estimated separately in age-stratified logistic regression models (<50y and ≥50y). P-values for interaction (risk score × Age) were obtained from separate logistic regression models that included the risk score, age (treated as a continuous variable), and their interaction term; the interaction test assesses statistical heterogeneity of the risk-score effect by age.”
- The Gail model requires the number of prior breast biopsies for calculation. As most participants in the study population have missing values for this variable, clarification is needed on how this was handled. Could this missingness have influenced the performance of the Gail model? An additional explanation would be helpful.
Our response:
We thank the reviewer for raising these important points. In our primary analysis, missing values for Gail model variables were assigned to the lowest risk (reference) category, consistent with prior validation studies of the Gail model. Because information on the number of prior breast biopsies was unavailable for the majority of participants (94% of Europeans and 100% of Asians), we did not perform multiple imputation for this variable, as any imputation would be driven largely by model assumptions rather than observed data.
We performed additional sensitivity analyses. In Methods:
“Handling of missing data and sensitivity analyses
Information on the number of prior breast biopsies was unavailable for the majority of participants (94% of included Europeans and 100% of included Asians). Because this variable was almost entirely missing, multiple imputation was not performed, as imputed values would have been determined primarily by model assumptions rather than observed data. For the primary analysis, missing values for all Gail model variables were assigned to the reference (lowest-risk) category, consistent with prior validation studies of the model.
To assess the potential influence of missing data on model performance, a sensitivity analysis was conducted in which missing values were instead assigned to the highest-risk category for each variable (age at menarche, age at first live birth, number of first-degree relatives with breast cancer, and number of prior breast biopsies). Five-year absolute risk was calculated using the R package BCRA, and the model’s discriminatory ability for invasive breast cancer was evaluated using the area under the receiver operating characteristic curve (AUC) with 95% confidence intervals.”
In results:
“In a sensitivity analysis assuming missing values corresponded to the highest risk category, the Gail model’s discriminatory ability changed notably among European women. (Supplementary Table 9, Supplementary Figure 4) For invasive disease, the AUC for the full model increased from 0.493 (0.487-0.499) to 0.627 (0.621-0.632) and for women <50 years and from 0.517 (0.514-0.520) to 0.561 (0.557-0.564) for those ≥50 years, suggesting that the large proportion of missing data, particularly for family history and biopsy variables, may have led to underestimation of model discrimination in the original analysis. Among Asian women, AUCs were similar between the two approaches (<50 years: 0.523 [0.511-0.535] vs. 0.507 [0.494-0.519]; ≥50 years: 0.554 [0.543-0.566] vs. 0.531 [0.520-0.543]).
In contrast, for DCIS, the Gail model showed minimal change or a reduction in AUCs when missing data were assigned to the high-risk category (Supplementary Table 10, Supplementary Figure 5). Among European women, the AUC for the full model was 0.521 (0.509-0.533) compared with 0.610 (0.597-0.622) in the original model for those <50 years, and 0.525 (0.518-0.531) vs. 0.519 (0.512-0.526) for those ≥50 years. Among Asian women, AUCs were similar or modestly higher under the high-risk assumption (0.589 (0.560-0.619) vs. 0.533 (0.505- 0.562) for <50 years; 0.599 (0.571-0.627) vs. 0.542 (0.513-0.572) for ≥50 years).”
As a limitation in Discussion:
“The high proportion of missing data for key Gail model variables represents an important limitation, particularly among European participants. Sensitivity analyses indicated that the model’s discrimination for invasive breast cancer was more affected by assumptions about missing data than for DCIS, likely due to the heavy weighting of family history and biopsy variables in the Gail model. The larger changes in AUC observed among Europeans suggest that missingness in these predictors contributed to greater uncertainty in model performance.”
- Potential drivers of Gail risk could be explored by including each factor in the logistic regression model. The methods section does not provide information on missing data handling. It may be useful to discuss possible imputation methods, particularly for missing family history variables.
Our response:
From the sensitivity analyses described above:
- For statistical comparisons, please consider the use of Fisher’s exact test where appropriate.
Our response:
We thank the reviewer for this suggestion. In Supplementary Tables 4-8, for variables with complete or near-complete missingness, such as the number of prior breast biopsies (missing for 94-100% of participants), Fisher’s exact test or any other statistical comparison cannot be meaningfully performed because there are no observed data to construct contingency tables. For variables with sufficient observed data, chi-square tests were used, and Fisher’s exact test was considered but not required given the large sample size. We therefore retained chi-square tests for the main analyses, as results would not change using Fisher’s exact test.
- To more directly compare the effects of PRS and Gail, it may be informative to include both indices simultaneously in the regression model, as they have been studied as independent risk factors.
Our response:
We have added the additional results to Table 2 and Supplementary Table 11.
Minor Comments
- The function “pnorm” referenced in the manuscript is an R function, not a package. Please revise accordingly.
Our response:
We have revised the text to:
“In brief, an individual's PRS percentile was obtained from the standardised PRS using the “pnorm” function in R.”
- The manuscript would benefit from additional editorial revision to improve readability and clarity.
Our response:
We thank the reviewer for this suggestion. As the comment is general and does not indicate specific sections, we have carefully reviewed the manuscript and made editorial adjustments where appropriate to improve readability and flow.
Reviewer 4 Report
Comments and Suggestions for Authors
In this article, the authors presented a comprehensive evaluation of polygenic risk scores and the Gail model for breast cancer risk stratification. They showed that PRS generally outperformed the Gail model. While the study leverages a large dataset and provides valuable insights into population-specific performance differences, I still have some concerns.
- The analysis on the excluded patients revealed systematic differences between included and excluded patients. Based on this observation, simply excluding patients assumes a MCAR missing mechanism. I would suggest performing proper missing data approaches such as multiple imputations or various NMAR methods and conduct sensitivity analysis to check if the exclusion biased their results.
- The BCRA automatically set missing values to baseline category. Again, this shoud be evaluated by properly handling the missing data.
- The authors stated that “recalibration or adjustment will be required before clinical implementation” and I think this is well supported by Figure 2. Can the authores provide specific recalibration approaches and demonstrate their performance? Or at least discuss what level of miscalibration would be acceptable?
Author Response
In this article, the authors presented a comprehensive evaluation of polygenic risk scores and the Gail model for breast cancer risk stratification. They showed that PRS generally outperformed the Gail model. While the study leverages a large dataset and provides valuable insights into population-specific performance differences, I still have some concerns.
- The analysis on the excluded patients revealed systematic differences between included and excluded patients. Based on this observation, simply excluding patients assumes a MCAR missing mechanism. I would suggest performing proper missing data approaches such as multiple imputations or various NMAR methods and conduct sensitivity analysis to check if the exclusion biased their results.
Our response:
We thank the reviewer for raising this important point. We acknowledge that exclusion of patients with missing values in key variables assumes that the data are missing completely at random (MCAR), which may not strictly hold. However, for some variables, such as the number of prior breast biopsies, missingness was extreme (94–100%), leaving virtually no observed data to inform imputation models. Under these circumstances, multiple imputation would be unreliable, as any imputed values would be driven primarily by model assumptions rather than observed data.
For other variables with more moderate missingness, such as first-degree family history (missing for 27% of Europeans and 8% of Asians), we evaluated potential bias using sensitivity (bounding) analyses. In these analyses, missing values were assigned the non-baseline categories to assess the maximum possible impact on model estimates and discrimination.
We have clarified this approach added Supplementary Tables 9-10 and Supplementary Figures 4-5) showing the sensitivity analyses. While formal NMAR approaches could theoretically be applied, they require strong assumptions. We have added a limitation in the discussion.
We performed additional sensitivity analyses. In Methods:
“Handling of missing data and sensitivity analyses
Information on the number of prior breast biopsies was unavailable for the majority of participants (94% of included Europeans and 100% of included Asians). Because this variable was almost entirely missing, multiple imputation was not performed, as imputed values would have been determined primarily by model assumptions rather than observed data. For the primary analysis, missing values for all Gail model variables were assigned to the reference (lowest-risk) category, consistent with prior validation studies of the model.
To assess the potential influence of missing data on model performance, a sensitivity analysis was conducted in which missing values were instead assigned to the highest-risk category for each variable (age at menarche, age at first live birth, number of first-degree relatives with breast cancer, and number of prior breast biopsies). Five-year absolute risk was calculated using the R package BCRA, and the model’s discriminatory ability for invasive breast cancer was evaluated using the area under the receiver operating characteristic curve (AUC) with 95% confidence intervals.”
In results:
“In a sensitivity analysis assuming missing values corresponded to the highest risk category, the Gail model’s discriminatory ability changed notably among European women. (Supplementary Table 9, Supplementary Figure 4) For invasive disease, the AUC for the full model increased from 0.493 (0.487-0.499) to 0.627 (0.621-0.632) and for women <50 years and from 0.517 (0.514-0.520) to 0.561 (0.557-0.564) for those ≥50 years, suggesting that the large proportion of missing data, particularly for family history and biopsy variables, may have led to underestimation of model discrimination in the original analysis. Among Asian women, AUCs were similar between the two approaches (<50 years: 0.523 [0.511-0.535] vs. 0.507 [0.494-0.519]; ≥50 years: 0.554 [0.543-0.566] vs. 0.531 [0.520-0.543]).
In contrast, for DCIS, the Gail model showed minimal change or a reduction in AUCs when missing data were assigned to the high-risk category (Supplementary Table 10, Supplementary Figure 5). Among European women, the AUC for the full model was 0.521 (0.509-0.533) compared with 0.610 (0.597-0.622) in the original model for those <50 years, and 0.525 (0.518-0.531) vs. 0.519 (0.512-0.526) for those ≥50 years. Among Asian women, AUCs were similar or modestly higher under the high-risk assumption (0.589 (0.560-0.619) vs. 0.533 (0.505- 0.562) for <50 years; 0.599 (0.571-0.627) vs. 0.542 (0.513-0.572) for ≥50 years).”
As a limitation in Discussion:
“The high proportion of missing data for key Gail model variables represents an important limitation, particularly among European participants. Sensitivity analyses indicated that the model’s discrimination for invasive breast cancer was more affected by assumptions about missing data than for DCIS, likely due to the heavy weighting of family history and biopsy variables in the Gail model. The larger changes in AUC observed among Europeans suggest that missingness in these predictors contributed to greater uncertainty in model performance.”
2. The BCRA automatically set missing values to baseline category. Again, this shoud be evaluated by properly handling the missing data.
Our response:
From the sensitivity analyses described above:
3. The authors stated that “recalibration or adjustment will be required before clinical implementation” and I think this is well supported by Figure 2. Can the authores provide specific recalibration approaches and demonstrate their performance? Or at least discuss what level of miscalibration would be acceptable?
Our response:
We have revised the Discussion to clarify the implications of miscalibration and potential recalibration approaches. This revision highlights the need for recalibration while acknowledging that acceptable thresholds depend on the clinical context:
“In practice, a well-calibrated model typically has a calibration slope close to 1 and an intercept near 0. In the absence of a universally accepted numerical threshold for deviations, it is important to consider miscalibration as clinically meaningful if it leads to over- or underestimation of risk that could influence clinical decision-making (34). Therefore, in real-world applications, recalibration or adjustment protocols, such as updating baseline incidence rate or performing logistic recalibration, will be necessary to ensure accurate absolute risk predictions before clinical implementation.”
Round 2
Reviewer 1 Report
Comments and Suggestions for Authors
Thank you for the revision.
Page 11: "Excluded Europeans differed significantly from those included regarding age at first full-term pregnancy, and missing data on age at menarche was more common among excluded
Europeans." Please mention what the "differed significantly" means.
Reviewer 4 Report
Comments and Suggestions for Authors
In this revision, the authors have addresses my previously raised issues. I have no further concerns.